# Depth Video-Based Secondary Action Recognition in Vehicles via Convolutional Neural Network and Bidirectional Long Short-Term Memory with Spatial Enhanced Attention Mechanism

**DOI:** 10.3390/s24206604

**Published:** 2024-10-13

**Authors:** Weirong Shao, Mondher Bouazizi, Ohtuski Tomoaki

**Affiliations:** 1Graduate School of Science and Technology, Keio University, Yokohama 223-8522, Japan; weirong0430@keio.jp; 2Faculty of Science and Technology, Keio University, Yokohama 223-8522, Japan; bouazizi@ohtsuki.ics.keio.ac.jp

**Keywords:** action recognition, deep learning, attention mechanism, depth sensor

## Abstract

Secondary actions in vehicles are activities that drivers engage in while driving that are not directly related to the primary task of operating the vehicle. Secondary Action Recognition (SAR) in drivers is vital for enhancing road safety and minimizing accidents related to distracted driving. It also plays an important part in modern car driving systems such as Advanced Driving Assistance Systems (ADASs), as it helps identify distractions and predict the driver’s intent. Traditional methods of action recognition in vehicles mostly rely on RGB videos, which can be significantly impacted by external conditions such as low light levels. In this research, we introduce a novel method for SAR. Our approach utilizes depth-video data obtained from a depth sensor located in a vehicle. Our methodology leverages the Convolutional Neural Network (CNN), which is enhanced by the Spatial Enhanced Attention Mechanism (SEAM) and combined with Bidirectional Long Short-Term Memory (Bi-LSTM) networks. This method significantly enhances action recognition ability in depth videos by improving both the spatial and temporal aspects. We conduct experiments using K-fold cross validation, and the experimental results show that on the public benchmark dataset Drive&Act, our proposed method shows significant improvement in SAR compared to the state-of-the-art methods, reaching an accuracy of about 84% in SAR in depth videos.

## 1. Introduction

Secondary Action Recognition (SAR) in vehicles is important as it enhances driving safety, in-car comfort, and overall driving experience. By monitoring secondary actions of both drivers and passengers, such as interacting with mobile phones, eating, or using electronic devices like laptops, SAR helps reduce distractions, provides fatigue warnings, and automatically adjusts the in-car environment [1,2]. This technology also provides valuable data support for autonomous driving, aiding in understanding passenger behavior patterns, improving emergency response capabilities, and enhancing human–vehicle interaction to deliver more intelligent and personalized services.

Most existing SAR technologies primarily depend on RGB images or videos, while these methods are widely used and have proven to be effective in many scenarios [3], they are not without significant limitations. One of the major drawbacks of relying on RGB images or videos is their susceptibility to environmental factors [4]. Conditions such as varying light levels and adverse weather can severely impact the quality of the data, thereby reducing the accuracy and reliability of the recognition process.

Moreover, the use of RGB images and video data raises substantial privacy concerns. Capturing detailed visual information about individuals can lead to potential misuse of data, infringing on personal privacy rights [5]. Especially now that cars are getting smarter, private information collected by in-vehicle cameras will likely be transmitted to public networks such as the cloud, making it vulnerable to personal information leakage. As a result, we require more reliable and private action recognition technologies.

Our research utilizes a depth sensor dataset extracted using Time-of-Flight (ToF) technology to address these challenges. It overcomes the limitations of traditional action recognition methods that rely on RGB image or video data. The ToF uses the time it takes for light to travel from the source to the object and back to the sensor to generate depth information [6]. Compared to traditional RGB cameras, ToF technology is less sensitive to these variations [6]. Because it relies on measuring the ToF of light rather than color or brightness, ToF maintains high accuracy in a variety of environments, including low-light or high-glare situations. Furthermore, unlike RGB images and videos that capture detailed visual information that may be used to recognize actions, the depth videos generated by ToF provide only spatial information and do not show identifiable or privacy features. The depth video generated by this method will contain some noise [6], so we use a GAN-based method [7] for noise reduction in depth videos.

In addition to employing the depth sensor, this paper introduces a novel comprehensive framework for SAR, depicted in Figure 1. Our proposed model integrates Convolutional Neural Network (CNN) and Bidirectional Long Short-Term Memory (Bi-LSTM) networks, further enhanced by a Spatial Enhanced Attention Mechanism (SEAM). The integration of multiple technologies forms a powerful video analysis framework suitable for action recognition tasks. Each technology uniquely handles different aspects of video processing, thereby improving the overall performance and accuracy of the model.

CNNs play a key role in extracting high-level spatial features in video frames. They can identify complex details and patterns, which are essential for understanding the visual composition of each frame. A SEAM enhances the capabilities of CNNs by guiding the model to focus on key areas within the frame. This attention mechanism ensures that the model does not treat all areas in the frame equally but rather prioritizes areas that are more relevant to the task, such as areas where actions occur. The SEAM helps the model distinguish between major and minor actions within a frame, which is crucial when accurately recognizing actions because subtle features often have important meanings.

Bi-LSTM extends the application of this framework by processing the characteristics of dynamic videos. Compared to traditional LSTM, it extends this process by processing information from the future to the past, providing a comprehensive temporal context [8]. This bidirectional processing allows the model to understand the previous and next frames, providing a more comprehensive view of temporal dependencies and action transitions. This understanding is essential for accurately predicting and recognizing actions because it ensures that the temporal flow of events is correctly interpreted.

By integrating CNN, SEAM, and Bi-LSTM, not only are the spatial accuracy and understanding of temporal context improved but also the action recognition system is significantly improved by focusing attention. This multifaceted integrated approach is critical for tasks that require a precise understanding of continuous motion in videos, making the model highly accurate for real-world applications. This combination of techniques demonstrates the power of combining specialized techniques to solve real-world action recognition problems.

To summarize, our main contributions are as follows:We utilize the Real Enhanced Super-Resolution Generative Adversarial Network (real-ESRGAN) [7] to effectively reduce noise in depth videos. This approach significantly improves the visual clarity and quality of the depth images, making them more suitable for detailed analysis and further processing.We propose the CNN-SEAM+Bi-LSTM model, which exploits the spatial strengths of CNN augmented by SEAM to intensify the focus on critical segments within the video frames. This spatial enhancement is complemented by the temporal learning capabilities of Bi-LSTM networks, which analyze and interpret the temporal sequences across video frames for more accurate predictions and recognitions.We demonstrate the effectiveness of our proposed CNN-SEAM+Bi-LSTM model for Secondary Action Recognition on the published Drive&Act [9] dataset using cross validation. Furthermore, on this benchmark dataset, we perform a comprehensive comparison with the state-of-the-art model.We conducted ablation experiments of SAR in depth videos and RGB videos in different lighting conditions using the CNN-SEAM+Bi-LSTM model. Our experiments demonstrated that this model is better than conventional RGB videos for SAR in depth videos.

## 2. Related Work

### 2.1. Depth Sensor in Human Action Recognition

The evolution of human action recognition (HAR) leveraging depth sensors has seen significant contributions from various studies exploring the synergy between sensor technologies and deep learning methods. AC Popescu et al. [10] proposed the HAR system that utilizes a Neural Architecture Search (NAS) to create an efficient and customizable solution. It integrates information from multiple sources, including RGB and depth data, skeleton, and context objects using 2D CNNs. Fusion mechanisms combine the outputs of these networks into a single array of class scores. Basak et al. [11] introduced DSwarm-Net, employing deep learning and a swarm intelligence metaheuristic for HAR with 3D skeleton data. Four features are extracted from skeletal data (Distance, Distance Velocity, Angle, and Angle Velocity) and encoded into images for simplified classification, achieving competitive results on NTU RGB-D datasets [12]. Batool et al. [13] proposed a surveillance system using fused sensors, focusing on RGB-D data. They introduced a modified K-Ary entropy classifier algorithm to improve action recognition, addressing challenges such as false alarms and depth information loss. The model extracts features from RGB and depth images using 2.5D cloud point modeling and ridge extraction, optimized using the PBIL algorithm. Yu et al. [14] presented MMNet, a model-based multimodal network for RGB-D HAR. Ahn et al. [15] proposed a Spatio-TemporAl cRoss (STAR)-transformer, which combines spatial–temporal videos and skeleton features into recognizable vectors using a novel encoding scheme. Employing full spatial–temporal attention and introducing zigzag and binary attention modules for efficient feature representation, experimental results on the NTU-RGB-D datasets demonstrate promising improvement in HAR under depth sensors.

### 2.2. Actions Recognition in Vehicles

Traditional action recognition in vehicle models from videos typically relied on manual feature descriptors [16,17,18,19,20]. However, recent advances have seen end-to-end deep learning models, particularly CNNs, become the preferred approach for recognizing individual actions in vehicles [21,22]. Xing et al. [23] developed a CNN-based system that acknowledges seven common driving actions, including normal driving, checking mirrors, using the in-car radio, texting, and answering phone calls. They employed transfer learning to fine-tune three pretrained CNN models—AlexNet [24], GoogleNet [25], and ResNet50 [26]—achieving accuracies of 81.6%, 78.6%, and 74.9%, respectively. Saranya et al. [27] utilized the Gaussian Mixture Model (GMM) for image segmentation as an input to their models while training CNNs for binary classification to ascertain driver distraction. Nel et al. [28] explored the application of ResNet with three-dimensional (3D) kernels to identify distracted driver behaviors. Peng et al. [29] leveraged visual transformers and an augmented feature distribution calibration module for enhancing driver action recognition. Alina Roitberg et al. [30] introduced a novel method named Calibrated Action Recognition with Input Guidance (CARING), employing an additional neural network to adjust confidence levels based on video input. Liu et al. [31] and Tanama et al. [32] innovated lightweight action recognition models using knowledge distillation techniques. Liu et al. [33] presented CEAM-YOLOv7, an enhanced YOLOv7 model incorporating a Global Attention Mechanism (GAM) and simplified convolution computation, achieving a 20.26% higher mean Average Precision (mAP) than the original YOLOv7 for detecting driver distraction behavior.

### 2.3. Attention Mechanism in Action Recognition

The attention mechanism in the context of action recognition in video sequences is a powerful tool in machine learning, especially in the field of computer vision. It allows models to focus on specific parts of a video that are more relevant for understanding the ongoing action, improving both the accuracy and the efficiency of action recognition tasks [34,35]. Wang et al. [36] and Xu et al. [37] utilized channel attention mechanisms integrated within convolutional neural networks, augmented by sensor data, to refine the processing of spatial features relevant to specific actions. Similarly, Mekruksavanich et al. [38] applied a hybrid approach that leverages channel attention to boost the performance of sensor-based action recognition systems. In the context of collective action recognition, which involves multiple individuals interacting within a scene, Lu et al. [39] developed a sophisticated model that employs spatio-temporal attention mechanisms. Their model not only focuses on individual actions by analyzing deep RGB features and human articulated poses but also captures the dynamics of group interactions, significantly enhancing the understanding of collective behaviors in complex visual scenes. Furthermore, addressing safety-critical applications such as driving, Imen Jegham et al. [40] introduced a novel hard attention network designed for Driver Action Recognition (DAR). This network is designed to pinpoint and analyze local salient regions that indicate the driver’s action, effectively distinguishing between safe driving and distracted behaviors. Their model demonstrated remarkable accuracy, achieving up to 95.83% in recognizing safe driving behaviors and up to 99.07% in detecting distractions, thereby offering a promising solution for enhancing road safety through advanced monitoring of driver attentiveness.

## 3. System Description

### 3.1. Overall System Description

The workflow of our experiment is depicted in Figure 2. This system targets SAR applications, utilizing the Drive&Act dataset [9]. Firstly, based on its annotations, we capture various categories of action clips related to SAR from depth videos in the Drive&Act dataset. The data distribution of the Drive&Act dataset is not even, and during data processing, we performed data augmentation on the dataset to balance the action category proportions, reduce the bias of the model, and enhance the robustness of the model. In addition, we use super-resolution to enhance the data quality, remove the noise from the image, and improve the image clarity.

Secondly, we extract video frames for spatial feature analysis. In this step, a CNN is used in conjunction with a SEAM, which is modified by Spatial Group-wise Enhance (SGE) [41], to more effectively enhance feature representation. The system’s architecture comprises multiple CNN layers, and a SEAM is integrated at each layer. This integration ensures that spatial features are consistently and effectively enhanced across the entire network. By applying a SEAM at every layer, we aim to capture intricate spatial details and improve the overall feature representation.

For the analysis of temporal features, Bi-LSTMs are employed. Bi-LSTMs are capable of learning information from both forward and backward directions, which enhances the model’s ability to understand the temporal dynamics of actions within video sequences. After feature extraction is complete, all spatial and temporal features are integrated and normalized through a softmax function to produce the final action detection and recognition results.

### 3.2. Data Preprocessing

#### 3.2.1. Data Augmentation

The Drive&Act dataset [9] initially includes thirty-four categories of fine-grained actions. However, not all actions are pertinent to secondary vehicle interactions such as opening and closing doors, entering and exiting the car, and so on. As such, we extracted and categorized twenty-five actions that relate directly to secondary actions. The dataset was originally highly undistributed, with some actions disproportionately underrepresented. To address the uneven distribution of data, we employed the data augmentation (DA) technique. Through observation, we found that the types of some actions were similar, such as ‘Putting on Sunglasses’ and ‘Taking off Sunglasses’, ‘Closing Laptop’ and ‘Opening Laptop’, ‘Putting Laptop into Backpack’ and ‘Opening Backpack’; these data have similar action gestures and fewer samples of individual actions, so we combined these actions into one label each: ‘Interacting with Backpack’, ‘Interacting with Laptop’, and ‘Interacting with Sunglasses’.

Further, for other actions with fewer samples, such as ‘Preparing Food’ and ‘Looking or Moving Around (e.g., searching)’, we performed DA techniques including random shifts, cropping, and vertical and horizontal flips. Examples of these can be seen in Figure 3, ensuring that the overall data distribution did not become extremely unbalanced. The exact number of data distributions before and after DA and the names of action types are shown in Figure 4. The details of the parameters of the video dataset after DA are shown in Table 1.

#### 3.2.2. Super-Resolution

Noise is an inevitable issue in depth videos, including both depth noise and temporal noise, which can affect the final accuracy of models. The super-resolution (SR) method using the Real-ESRGAN technique [7] effectively removes noise from depth videos, improving the overall video quality. This technique employs several key methods, including deep learning denoising, multi-scale feature extraction, Generative Adversarial Networks (GANs) [42], and temporal consistency.

Firstly, deep learning denoising involves training a neural network model to identify and remove noise while preserving important depth information. Secondly, multi-scale feature extraction enables the model to accurately identify and eliminate various types of noise by extracting features at different scales. Additionally, GANs improve the realism and noise-free quality of depth maps through adversarial training between generators and discriminators. Lastly, maintaining temporal consistency between frames during the denoising process helps prevent flickering or jumping in the denoised video, ensuring the smoothness and stability of the video.

In summary, while noise in depth videos is complex and unpredictable, the application of Real-ESRGAN can effectively mitigate these issues, improving the overall quality and usability of the video. Examples of depth video before and after SR are illustrated in Figure 5 and Figure 6. Detailed SR enlargements are shown in Figure 7. The resolution of each depth video went from the original 512×424 to 1536×1272 after SR.

### 3.3. Detailed Model Description for SAR

Our proposed model is a deep learning framework designed for SAR, integrating time-distributed convolutional layers, SEAMs, and Bi-LSTM. In the overall structure of the model, time-distributed convolutional layers first process each frame of the video, gradually increasing the number of filters (16, 32, 64) while maintaining the frames’ dimensions through ReLU activation functions and consistent padding. Following each convolutional layer, a spatial group enhancement layer is equipped to improve the quality of the feature maps by enhancing the expression of features within the group. To reduce the model’s sensitivity to data noise and decrease overfitting, max pooling and dropout layers are incorporated.

After the extraction of spatial features, a flattening layer transforms the convoluted feature maps into a one-dimensional vector, which is then processed by a Bi-LSTM layer. This bidirectional structure enables the learning of information from the context before and after the time series, capturing complex dynamic relationships. Finally, a fully connected layer maps the feature vector to the probability distribution of predefined categories, outputting through a softmax function to achieve precise classification of actions within the video sequence.

#### 3.3.1. CNN-SEAM Spatial Feature Extraction

A CNN is extensively used for image and video analysis, excelling in extracting detailed features from visual data. The process involves convolutional layers that apply filters to capture basic image features. These layers are followed by ReLU activation functions that help the network learn complex patterns. Pooling layers reduce data size and complexity while maintaining important features. Finally, fully connected layers aggregate these features to perform classification or regression tasks, allowing the CNN to efficiently analyze and interpret visual data [43].

Although the CNN performs well in image and video analysis, it suffers from some limitations such as unavoidable noise and the existence of similar patterns [41]. To address these issues, an attention mechanism is introduced, which enhances the adaptability and flexibility of the model by dynamically choosing to focus on specific parts of the data and assigning different weights to different features [34].

Nowadays, popular attention mechanisms mainly focus on channel attention mechanisms and spatial attention mechanisms [34]. Channel attention focuses on the importance of each channel in feature maps generated by CNNs [44]. Spatial attention, on the other hand, highlights regions of an image that are more important for processing [45]. In depth videos, the value of each pixel typically represents the relative distance to the camera. This data format is usually in grayscale, with only one channel, where the size of the pixel value indicates depth (typically distance). Therefore, the channel’s attention mechanism does not work to its best advantage. In our work, we apply a SEAM modified by Spatial Group-wise Enhance (SGE) [41].

In the SGE framework, channels of a feature map are typically divided into several groups. However, due to our dataset containing only one channel, we bypass the grouping stage and focus directly on the spatial enhanced attention mechanism (SEAM), as depicted in Figure 8. The input frame consists of a single-channel, convolutional feature map whose size is H×W. Each position is represented by a vector $\chi =\{x_{1},\cdots ,x_{m}\},$$x_{i}\in R$, *m* is the number of vectors in each feature map, and its value is H×W. Given that the features of the entire space are not dominated by noise, we apply global average pooling further to enhance the learning of important regional semantic features:(1)$$g=\frac{1}{m}\sum_{i=1}^{m}x_{i}.$$

We then measure the similarity between the global semantic features g and each local feature $x_{i}$ using the dot product:(2)$$c_{i}=g\cdot x_{i}.$$

This metric assumes that features closer in vector length and direction to g are more likely to obtain a larger initial coefficient.

To mitigate the potential bias in the magnitude of coefficients across samples, we normalize *c* as follows [46]:(3)$$\mu_{c}=\frac{1}{m}\sum_{i=1}^{m}c_{i},$$
(4)$$\sigma_{c}^{2}=\frac{1}{m}\sum_{i=1}^{m}{\left(c_{i}-\mu_{c}\right)}^{2},$$
(5)$${\hat{c}}_{i}=\frac{c_{i}-\mu_{c}}{\sqrt{\sigma_{c}^{2}+\epsilon }}.$$

Here, ${\hat{c}}_{i}$ is the normalized value of $c_{i}$, $\mu_{c}$ represents the mean of all $c_{j}$ values, and $\sigma_{c}^{2}$ is the variance. ϵ is a small constant added to prevent division by zero, typically a small positive number such as $10^{-8}$. This normalization ensures unbiased comparisons between different samples.

To maintain the identity transformation within the network, we introduce a scaling and shifting parameter pair, γ and β, for each coefficient ${\hat{c}}_{i}$:(6)$$a_{i}=\gamma {\hat{c}}_{i}+\beta .$$

Finally, to obtain the enhanced feature vector ${\hat{x}}_{i}$, the original $x_{i}$ is scaled by the generated importance coefficient $a_{i}$ through a sigmoid function gate σ(·):(7)$${\hat{x}}_{i}=x_{i}\cdot \sigma \left(a_{i}\right).$$

All enhanced features together form the resulting feature group $\hat{\chi }=\left\{{\hat{x}}_{1},\ldots ,{\hat{x}}_{m}\right\}$, where ${\hat{x}}_{i}\in R$ and *m* is equal to the value of H×W.

Compared to other attention mechanisms that increase the complexity of the model, this method only adds limited additional computations at each convolutional layer to assess the importance of features at each spatial location without adding any other convolutional layers or fully connected layers. Therefore, it does not increase the model’s size.

In our work, the model contains a total of four convolutional layers, each using a 3×3 convolutional kernel in conjunction with a ReLU activation function and zero padding, a configuration designed to extract features while keeping the spatial dimensions of the original frames unchanged to retain as much information as possible.

The first convolutional layer uses 16 filters mainly responsible for capturing the primary features of the image such as edges and textures. As the layers deepen, the number of filters increases, with the second layer using 32 filters, which not only deepens the recognition of shape and structure but also begins to touch on some intermediate visual features such as angles and contours. The 64 filters each in the third and fourth layers further deepen the network’s recognition capabilities, allowing the model to capture more complex features. This layer-by-layer feature abstraction from low to high level is key to understanding and analyzing video frames’ content.

In each convolutional layer, SEAM technology is incorporated, as depicted in Figure 9. Positioned after the convolution and activation function and before the pooling layer, the SEAM optimizes feature expression by selectively concentrating on the more informative regions within the feature maps. This targeted focus enhances the model’s sensitivity and accuracy in processing visual data.

#### 3.3.2. Temporal Feature Extraction

LSTM networks are an advanced type of Recurrent Neural Network (RNN) [47] designed to address the difficulties of learning long-term dependencies in sequence data. Unlike standard RNNs, LSTMs incorporate several gates (forget gate, input gate, cell state, and output gate) that control the flow of information, ensuring that the network can maintain and update its state over long sequences without losing context [47].

The forget gate decides what information should be discarded from the cell state, implemented by a sigmoid function, formulated as
(8)$$f_{t}=\sigma \left(W_{f}\cdot \left[h_{t-1},x_{t}\right]+b_{f}\right),$$
where $W_{f}$ is the weight matrix of the forget gate, $h_{t-1}$ is the output from the previous time step, $x_{t}$ is the input at the current time step, and $b_{f}$ is the bias associated with the forget gate.

The input gate decides which new information is to be stored in the cell state. This is performed in two parts: one part uses a sigmoid function to determine the degree of updating, formulated as
(9)$$i_{t}=\sigma \left(W_{i}\cdot \left[h_{t-1},x_{t}\right]+b_{i}\right),$$
while the second part is a tanh layer that generates candidate state values, formulated as
(10)$${\tilde{C}}_{t}=\tanh\left(W_{C}\cdot \left[h_{t-1},x_{t}\right]+b_{C}\right),$$
where $W_{i}$ and $W_{C}$ are the weight matrices for the input gate and the candidate state, respectively, and $b_{i}$ and $b_{C}$ are the biases for the input gate and the candidate state.

The cell state is updated by combining the output of the forget gate and the input gate, with the update formula being
(11)$$C_{t}=f_{t}⊙C_{t-1}+i_{t}⊙{\tilde{C}}_{t},$$
where ⊙ denotes the element-wise multiplication.

Finally, the output gate determines the output based on the current cell state (processed through a tanh function), with the formula being
(12)$$o_{t}=\sigma \left(W_{o}\cdot \left[h_{t-1},x_{t}\right]+b_{o}\right),$$
where $W_{o}$ is the weight matrix of the output gate and $b_{o}$ is the bias associated with the output gate. The final output is determined by
(13)$$h_{t}=o_{t}⊙\tanh\left(C_{t}\right).$$

This structural design allows LSTM to process and predict dependencies in long sequences effectively, making them a powerful tool for addressing various time series challenges. In our experiments, we use Bidirectional LSTM (Bi-LSTM). Bi-LSTM is a variant of LSTM that contains two LSTM layers, one dealing with the forward transfer of data (from the past to the future) and the other dealing with the backward transfer of data (from the future to the past). This structure allows the network to have past and future contextual information when making predictions.

## 4. Experimental Settings

### 4.1. Equipment

In our experiment, we use the depth videos obtained from Microsoft Kinect for XBox One; the specifications of this device are given in Table 2.

The technology of Microsoft Kinect for XBox One is Time of Flight (ToF). This is a technology for remote sensing that utilizes beams of infrared laser (IR) to determine the distance between the sensor and an object. Unlike traditional RGB cameras, which rely on visible light to produce images, depth sensors, such as those used in ToF, can operate effectively in total darkness. These sensors are designed to capture three-dimensional spatial information. Here is how they work: The system emits a short, precise laser pulse which, upon striking an object, is reflected to the sensor. The device then calculates the time delay between sending and receiving the laser pulse. Using the known speed of light, the sensor computes the distance to the object using the following formula:(14)$$\mathrm{Distance}=\frac{1}{2}\times c\times \Delta t,$$
where *c* denotes the speed of light and Δt denotes the time it takes for light to travel back and forth between a person and an object. The process is shown in Figure 10. By rapidly repeating this process, depth sensors can generate the depth information of the surveyed environment.

### 4.2. Dataset

In our experiment, we used the public dataset Drive&Act [9]. The dataset contains over 9.6 million frames of video spanning 12 h, documenting the behavior of individuals engaged in distracting actions while in both manual and autonomous driving modes. This comprehensive dataset captures footage from six perspectives, incorporating color, infrared, depth, and 3D body posture information. Each video has been intensively annotated with a hierarchical labeling system, covering 83 distinct categories. These data provide a valuable resource for understanding and analyzing driver attention and distraction patterns in different driving contexts.

Our experiment focuses on depth videos that contain the fine-grained actions obtained by Microsoft Kinect for XBox One in the Drive&Act dataset, as these actions predominantly relate to secondary actions performed by drivers while driving, such as interacting with phones, reading newspapers, drinking, eating, and so on. The camera angle is located in the upper-right corner. These tasks are particularly relevant for SAR. Examples of these fine-grained actions, obtained from various resolutions, are illustrated in Figure 11. The details of data distribution and parameters in our experiments are described in Figure 4 and in Table 1.

### 4.3. Experimental Configuration

For our project, the experiments’ configuration is in Table 3. The main training hyperparameters are shown in Table 4. Our experiments employ a patience of 15 epochs for early stopping, which helps prevent overfitting by halting training if no improvement is observed based on the loss or accuracy. It uses categorical cross entropy as the loss function, suitable for SAR which has multi-class classification tasks. Data shuffling is enabled to prevent the model from learning any order-based biases from the training data. Additionally, 20% of the data are reserved for validation to help monitor and prevent overfitting.

### 4.4. Evaluation Metrics

Given that varying experiments yield different models, it is essential to establish a robust metric. This aids in the selection of the most effective models from the entirety of the experimental outcomes. In our experiments, we used the accuracy score calculated according to Equation (Equation 15):(15)$$\mathrm{Accuracy}=\frac{\mathrm{TP}+\mathrm{TN}}{\mathrm{TP}+\mathrm{TN}+\mathrm{FP}+\mathrm{FN}}.$$

In this formula, TP (True Positives) refers to the correct positive predictions, and TN (True Negatives) refers to the correct identification of the negative class. FP (False Positives) occurs when the model incorrectly identifies a negative as a positive, and FN (False Negatives) occurs when the model fails to identify a positive. The accuracy metric calculates the proportion of true results (both positives and negatives) over the total number of evaluations, providing a straightforward measure of how effectively the model classifies multiple categories.

## 5. Experimental Results and Discussion

### 5.1. Model Evaluation with K-Fold Cross Validation

Cross validation is a statistical analysis method used to assess the generalization ability of a deep learning model. It is mainly implemented by splitting the dataset into multiple small subsets, which ensures that the model is not exposed to the data used for testing during training, thus providing a more accurate assessment of the model’s performance [48]. In our experiments, we employ a K-fold cross-validation approach, a specific form of cross validation [49]; we set k equal to 5, where the data are evenly divided into five equal-sized folds. The model undergoes the training and test process five times using the dataset after super-resolution (SR), each selecting a different fold as the test set (for evaluating the model) and the other four folds as the training set.

The results from our experiments are in Table 5, showing varying test losses and accuracies across five folds. The variation trend of losses and accuracies of test datasets is shown in Figure 12. Test loss ranged from the highest of 0.82 in Fold-2 to the lowest of 0.44 in Fold-5, indicating that the model had the best generalization in Fold-5 and the worst generalization in Fold-2. Correspondingly, accuracy varied from 79.05% in Fold-2 to 87.73% in Fold-5. The lowest accuracy correlates with the highest test loss, reinforcing that Fold-2 was the most challenging subset for the model. There may be noise in Fold-2 or overfitting of the model to these training data. In general, the average accuracy is 83.88% with a standard deviation of 2.96%, and the overall average test loss is 0.59 with a standard deviation of 0.13. This indicates that the model is relatively stable in overall performance and has a high average accuracy and low loss value.

The cumulative confusion matrix of cross validation is shown in Figure 13. The prediction accuracy for most actions is close to or above 80%, indicating that the model has high accuracy in recognizing actions. According to the cumulative confusion matrix, high-accuracy actions typically share common characteristics; for example, ‘Reading Magazines’, ‘Reading Newspaper’, and ‘Using Multimedia Display’ often have distinct and consistent motion patterns like stable hand positions and holding special objects, making it easier for models to learn and recognize these consistent patterns. However, the classification accuracy of some actions is relatively low. For instance, ‘Looking or Moving Around (e.g., searching)’ has only 54% accuracy, which is one of the worst-performing categories. This action involves a lot of body movement and the features are not clear enough, which makes it easy for the model to confuse it with other actions, such as ‘Unfastening Seat Belt’, ‘Putting on Jacket’, or ‘Handling an Object’, which also contain similar features to ‘Looking or Moving Around’.

### 5.2. Model Evaluation with DA and SR Techniques

In this section, we evaluate the performance of our SAR model using the initial data and after the application of data augmentation (DA) and super-resolution (SR). The results are depicted in Table 6. Initially, the overall performance showed an incremental improvement, starting from 69.27% in the original data, increasing to 74.36% after DA, and further rising to 83.88% after SR. This suggests that both processes significantly improve the quality of data or the effectiveness of subsequent analyses.

DA exhibits positive effects on the SAR. Notable increases in SAR accuracies are observed in actions such as ‘Interacting with Laptop’, ‘Interacting with Backpack’, and ‘Interacting with Sunglasses’. These actions combine related activities, where their accuracy jumps from 39.17% to 42.86%, from 60.72% to 81.58%, and from 20.71% to 75.00%, respectively. This demonstrates that integrating similar small samples of data into a larger sample can significantly improve accuracy. Additionally, actions with fewer samples, such as the ‘Pressing Automation Button’, also show significant improvements in accuracy following the application of DA techniques to increase training samples, with an accuracy improvement from 78.57% to 92.11%. These results underscore the effectiveness of DA in enriching the variability and quality of the training dataset, leading to improved model performance.

SR consistently enhances performance across almost all actions when compared to the augmented data. This improvement is particularly pronounced in actions with initially lower accuracy, such as ‘Interacting with Laptop’ and ‘Unfastening Seat Belt’, where accuracies increase significantly to 69.39% and 56.00%, respectively. This indicates the effectiveness of SR in enhancing feature extraction or recognition capabilities. Actions involving complex interactions or detailed movements, like ‘Interacting with Laptop’ and ‘Interacting with Sunglasses’, show substantial benefits from SR, with accuracies improving to 69.39% and 83.61%, respectively. Conversely, for simpler or more static actions such as ‘Sitting Still’ and ‘Reading Magazine’, the gains are less pronounced, suggesting a potential limit to how much these enhancement techniques can improve recognition in scenarios where the original actions are straightforward.

### 5.3. Model Performance Comparisons for SAR

We conducted experiments with and without the SEAM to verify that the SEAM is effective for SAR enhancement. The results are shown in Table 7. The accuracy is improved by 14.71% with the SEAM. This enhancement emphasizes the SEAM’s role in better capturing and emphasizing critical features of the video frames in the dataset, thereby boosting the model’s performance and reliability in SAR.

To evaluate the impact of the number of LSTM layers and the number of hidden units within each layer on the network’s performance, we conducted experiments with various configurations of both hidden units and LSTM layers and made a comparison with Bi-LSTM. The test accuracy is shown in Table 8. According to the result, the Bi-LSTM model excels in most settings compared to one-layer LSTM and two-layer LSTM with different units, particularly at 128 units where it reaches the highest accuracy of 83.88% across all the models and configurations tested. This suggests that the bidirectional approach, which utilizes both past and future context, is particularly effective in enhancing the accuracy of SAR. The trend across the models indicates that increasing the number of units initially improves performance up to 128 units, after which the benefits diminish due to overfitting. Therefore, in the temporal sequence part of our experiment, we use the Bi-LSTM layer with 128 hidden units.

In feature extraction for action recognition, the performance varies significantly when the model is configured with different numbers of CNN layers. Through our experiments, we explored feature extraction using three, four, and five CNN layers, the result is shown in Table 9. The initial two layers consist of 16 and 32 3×3 filters, respectively, while subsequent layers utilize 64 3×3 filters each. Each layer incorporates the ReLU activation function and employs ‘same’ padding to enhance nonlinearity, maintain the output size, and improve the effectiveness of feature extraction with the SEAM.

The experimental results demonstrate that configuring the model with a three-layer CNN achieves an accuracy of 81.12%. This outcome indicates that even a minimal number of layers can effectively perform time series analysis with Bi-LSTM layers, adequately capturing essential features. However, when the configuration is extended to four CNN layers, the accuracy significantly rises to 83.88%. Conversely, increasing the number of layers to five leads to a decrease in accuracy to 76.56%. This reduction is due to the additional layers inducing overfitting or rendering the model overly complex for optimal performance.

Next, we saved the training model and used that model to predict video actions using the test dataset (20% of each category’s actions). The prediction results and prediction probabilities are shown in Figure 14. As a result, all of the actions can be predicted, and the prediction probability can exceed 80%.

In the end, we conducted experiments to compare the proposed CNN-SEAM+Bi-LSTM model with other state-of-the-art methods on the benchmark Drive&Act dataset, as shown in Table 10. The results demonstrate that our proposed model, CNN-SEAM+Bi-LSTM, significantly outperforms all the other models, achieving the highest accuracy of 83.88%. This analysis highlights the advancements made by the CNN-SEAM+Bi-LSTM model in SAR, as evidenced by its superior performance on the benchmark Drive&Act dataset.

### 5.4. Ablation Experiments on RGB and Depth Datasets

To validate the effectiveness and stability of our proposed method, we conducted ablation experiments on the RGB dataset. These experiments were set up under the same conditions as described in Section V.C. We simulated different lighting conditions, as depicted in Figure 15, to observe the differential outcomes yielded by our method in each distinct lighting environment. We adjusted the brightness of the RGB video to 10% of the original brightness to simulate the interior of a vehicle in a dark environment. The result is shown in Table 11.

Overall, the accuracy of the depth video is 83.88%, which is higher than the 65.87% of the RGB video and the 58.36% of the RGB video in the dark condition. Specific analysis reveals that for most actions, the accuracy of the depth video is higher than that of the RGB video and RGB video in the dark condition; for example, ‘Eating’, ‘Fastening Seat Belt’, and ‘Drinking’ have accuracies of 91.06%, 72.17%, and 91.53% in the depth video, respectively. For some actions, the RGB video performs almost the same as the depth video, such as for ‘Talking on the Phone’ and ‘Using Multimedia Display’, which have accuracies of 92.56% and 97.32%, respectively. However, the accuracy of the RGB video significantly decreases under low-light conditions, demonstrating that such conditions greatly impact the performance of the RGB video model.

The higher accuracy of depth video can be attributed to several factors. Firstly, depth video is more robust to changes in lighting conditions and is less affected by changes in lighting, allowing it to maintain a high level of reliability in a variety of environments. Second, depth video provides rich spatial information, which is critical for distinguishing between different actions and helps the model understand the 3D structure of the scene and the relative positions of objects and body parts. In addition, depth video usually contains less background noise, which helps the model to focus on action-related features, especially when a SEAM is employed in the model, which is more capable of focusing on parts of the video frames that provide more useful information from the video frames, thus improving the recognition performance.

While our model demonstrates superior performance with depth videos compared to RGB videos under various conditions, there remain certain limitations to our approach. Firstly, high-quality depth sensors are often associated with high costs, potentially limiting their widespread adoption in consumer markets. Additionally, depth sensors can face challenges in accurately detecting certain types of objects or materials, such as transparent or highly reflective surfaces. These materials can disrupt the sensors’ infrared or laser signals, adversely affecting the depth perception capabilities of the system. This limitation may impact the accuracy and reliability of action recognition.

## 6. Additional Experiments

### 6.1. Model Performance on Other Datasets

To further validate the wide applicability and robustness of our model, we evaluated it on three additional datasets for action recognition: KTH [54], UCF50 [55], and UCF101 [56]. The KTH dataset is a video dataset used for human action recognition research, containing videos of 25 individuals performing six types of actions in different scenarios. The UCF50 dataset contains real-world action videos from 50 categories. UCF101 is an extended version of UCF50, comprising 101 action categories and covering a broader range of activities and scenarios. These datasets are common benchmark datasets in video action recognition and are widely used to evaluate the performance of video action recognition algorithms. The common point between the Drive&Act dataset and these three datasets is that they all record various types of single-person daily life actions, and the similarity between action categories is strong.

The results of our model for different datasets are presented in Table 12. The result of the accuracy of the model on the KTH dataset is 65.60%, which is the lowest accuracy of the three datasets, probably due to the smaller size and smaller number of categories in this dataset. In contrast, on the UCF50 and UCF101 datasets, which have more categories, the model demonstrates a higher accuracy rate of 89.24% and 85.06%, respectively. This indicates that the CNN-SEAM+Bi-LSTM model is more suitable for handling action recognition tasks with more categories and complexity. Overall, our model achieves an action recognition accuracy of over 65% in all three of the other datasets, indicating the broad feasibility and robustness of our model.

### 6.2. Performance of the Model in the Case of Vehicle Shaking

When driving a vehicle, due to the uneven smoothness of the road surface, the vehicle will inevitably shake, which causes the camera in the car to shake, and the displayed video will be blurred. This is a very common situation in real life. Because the public dataset Drive&Act that we used does not have video data in a similar situation, in order to test whether the model we generated can still have a high SAR accuracy when the video is shaking, we used CapCut 4.8.0 video processing software to analyze the depth video of Drive&Act and added ‘camera shake’ and ‘motion blur’ effects to simulate the shaking phenomenon during driving. The video screenshots before and after processing are shown in Figure 16. After we batch-generated depth videos with the added special effects, we used the previously trained model to detect the actions in the generated shaky videos. The detection results are shown in Table 13 below. It can be seen from the results of the detection accuracy that the effect of shaking does not significantly affect the accuracy of SAR. Compared with 83.88% without special effects, the accuracy only dropped to 81.34%. Therefore, it can be seen that in the actual driving process, even if there is shaking, our model can still have a high SAR accuracy.

## 7. Conclusions and Future Work

Our research addresses the limitations of traditional RGB videos in Secondary Action Recognition (SAR) in vehicles by using depth videos generated by Time-of-Flight (ToF) technology. We then designed a CNN-SEAM+Bi-LSTM model framework to effectively mitigate environmental variability and privacy concerns associated with traditional SAR techniques. The proposed CNN-SEAM+Bi-LSTM model exploits the spatio-temporal dynamics of the video data to enhance the model’s ability to discriminate actions. The integration of a SEAM further refines the model’s focus on the most informative regions of the video frames, which improves the accuracy and reliability of the action recognition. We evaluated the proposed model on the benchmark Drive&Act dataset and achieved competitive results with state-of-the-art methods.

However, some limitations in the current work are still to be addressed. We used video data where each action lasts only one to three seconds, so it remains to be seen whether our model can maintain high action recognition accuracy over longer durations. Additionally, our current work focuses solely on the SAR of a single person (the driver), but in real-life scenarios, the actions of passengers can also impact driving safety. In addition, the dataset used in this work is publicly available [9]. There is currently no dataset that contains data collected simultaneously using different sensors and devices such as ADAS, edge computing devices, audio sensors, etc. Collecting and building such a dataset is both time- and resource-consuming. In our future work, we plan to start collecting our own data and creating a multi-modal system that incorporates multiple sensors and devices that typically exist or can be added to commercial vehicles to enhance the system’s applicability and validate its effectiveness in more complex and realistic driving environments.

## Figures and Tables

**Figure 1 sensors-24-06604-f001:**
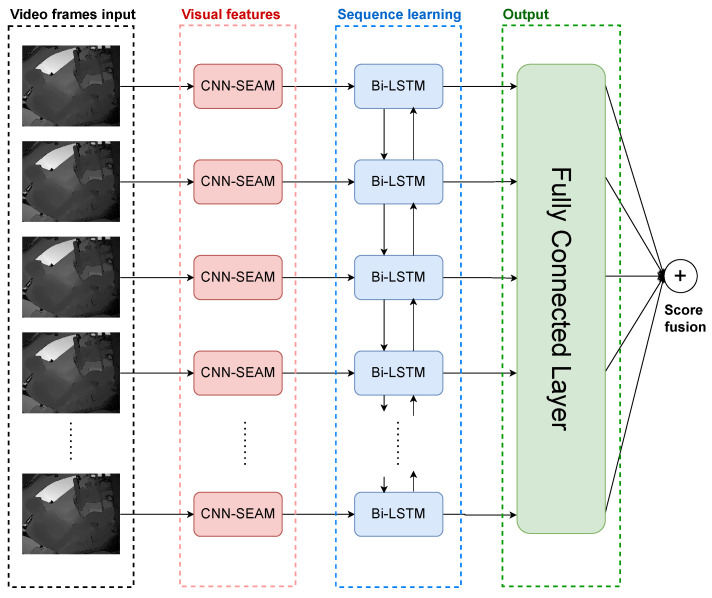
The structure of CNN-SEAM+Bi-LSTM model.

**Figure 2 sensors-24-06604-f002:**
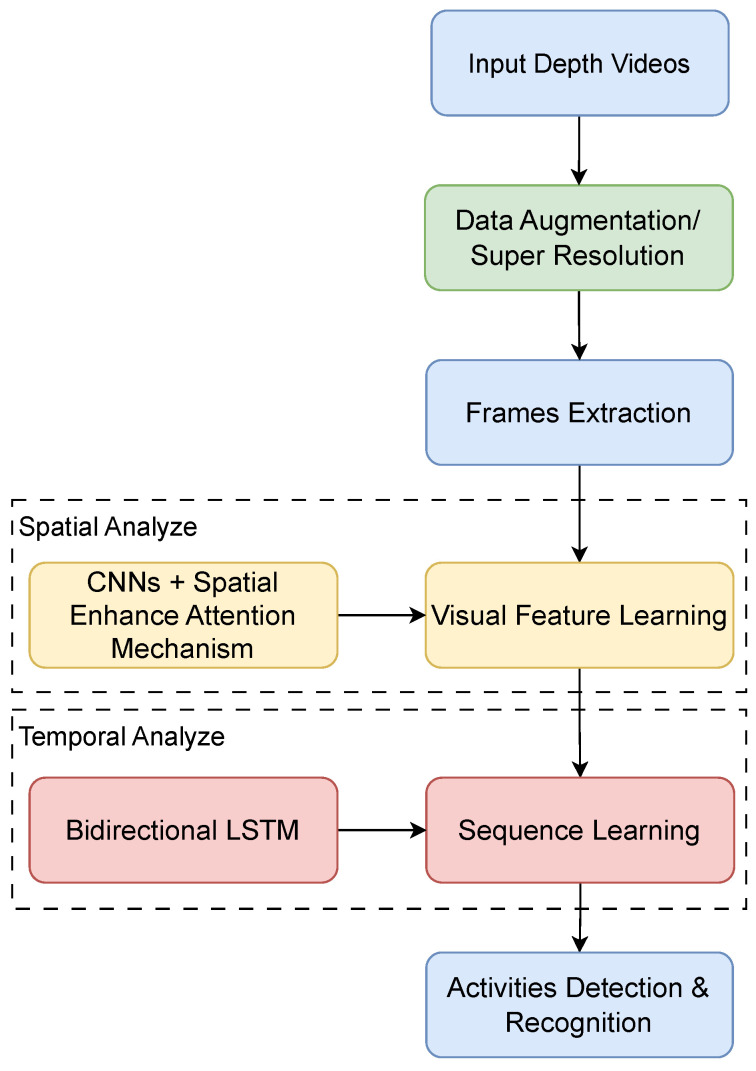
The overview of our experiment.

**Figure 3 sensors-24-06604-f003:**
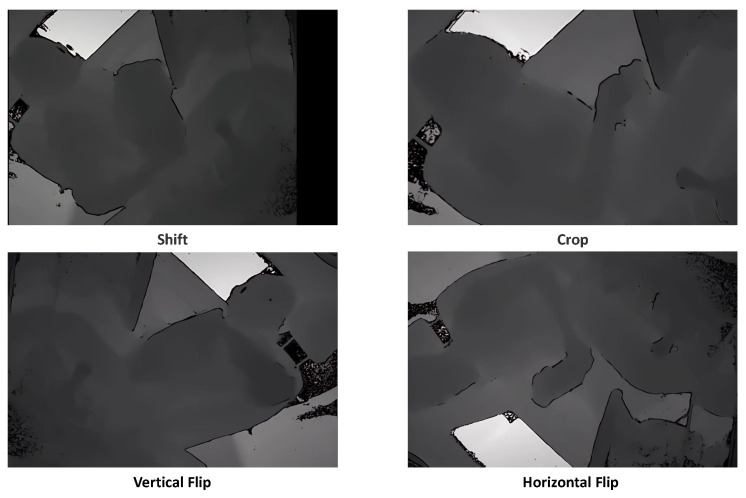
Examples of data augmentation.

**Figure 4 sensors-24-06604-f004:**
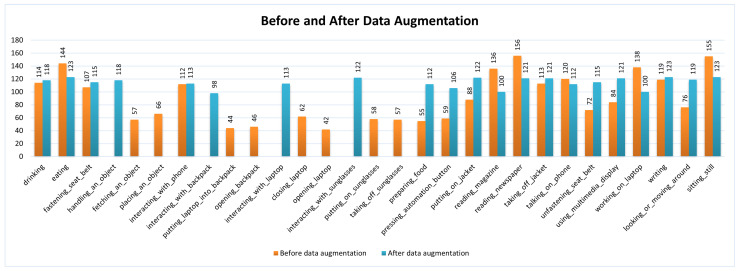
Data distribution.

**Figure 5 sensors-24-06604-f005:**
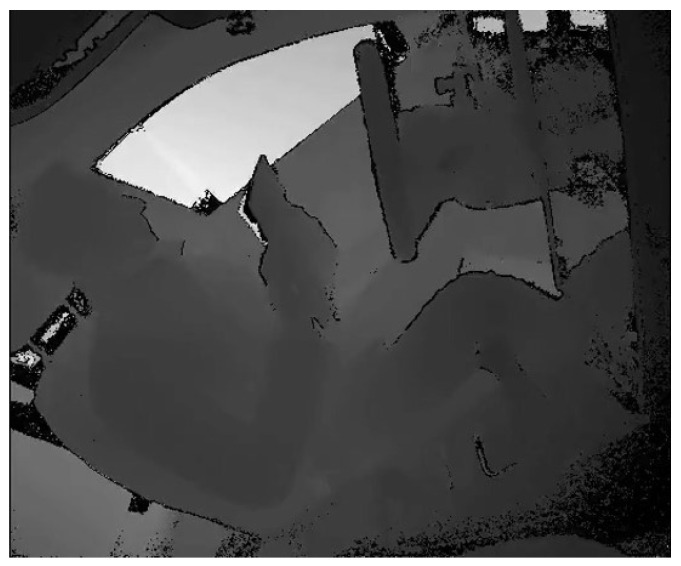
Depth video before SR.

**Figure 6 sensors-24-06604-f006:**
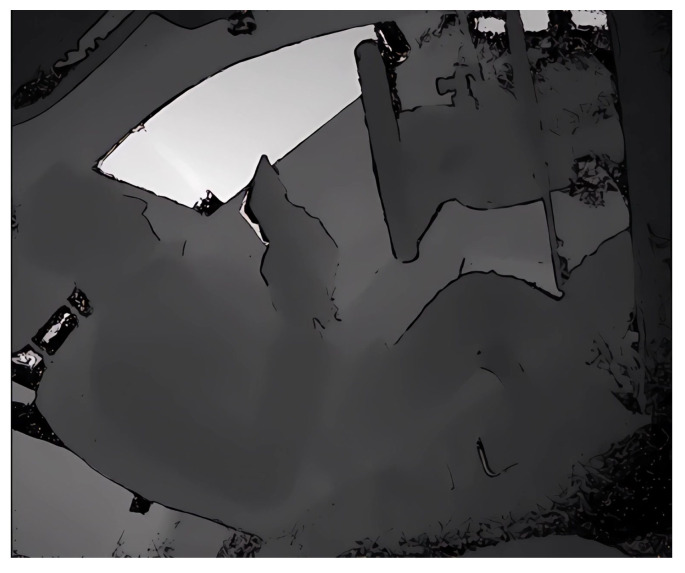
Depth video after SR.

**Figure 7 sensors-24-06604-f007:**
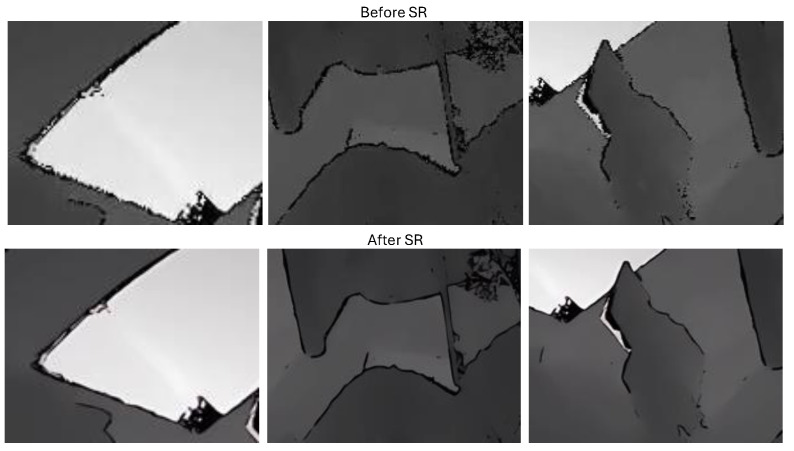
The detail of SR: The first row is the local zoom image before SR and the second row is the local zoom image after SR.

**Figure 8 sensors-24-06604-f008:**
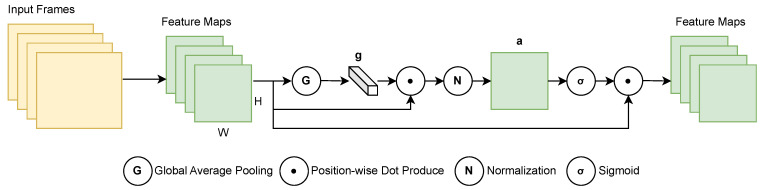
The structure of the SEAM.

**Figure 9 sensors-24-06604-f009:**
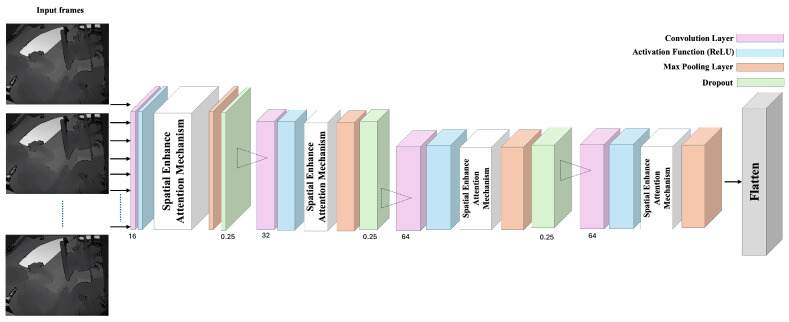
The structure of CNN-SEAM.

**Figure 10 sensors-24-06604-f010:**
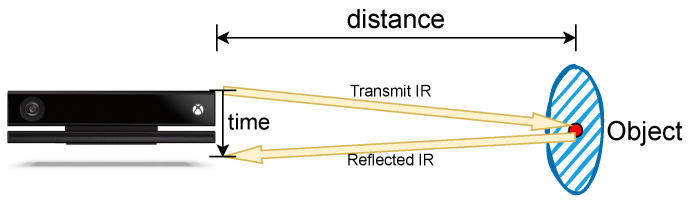
The working process of ToF.

**Figure 11 sensors-24-06604-f011:**
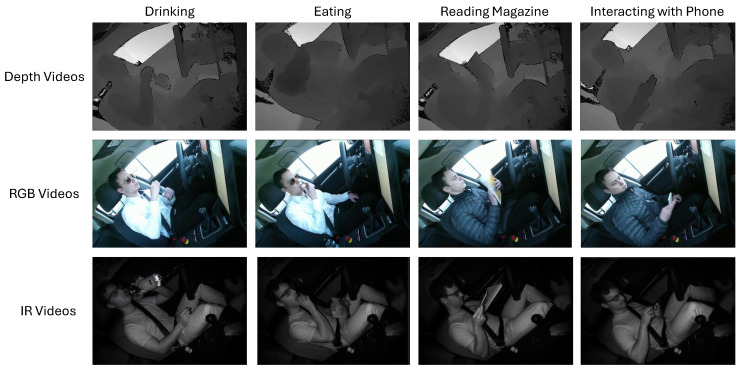
Examples of video data of Drive&Act: The first line consists of depth videos, the second line consists of RGB videos, and the third line consists of IR videos.

**Figure 12 sensors-24-06604-f012:**
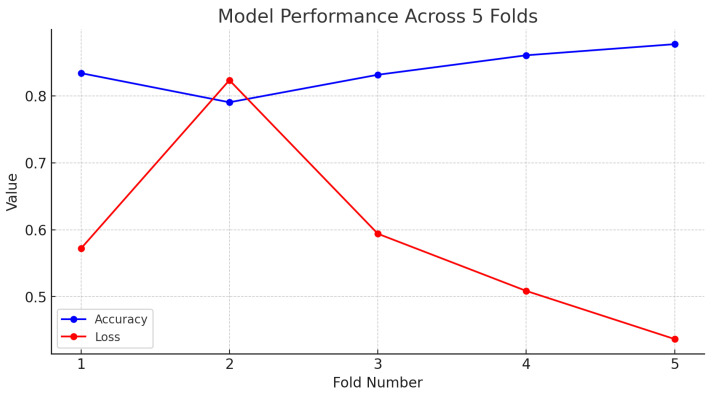
Model performance across 5 folds.

**Figure 13 sensors-24-06604-f013:**
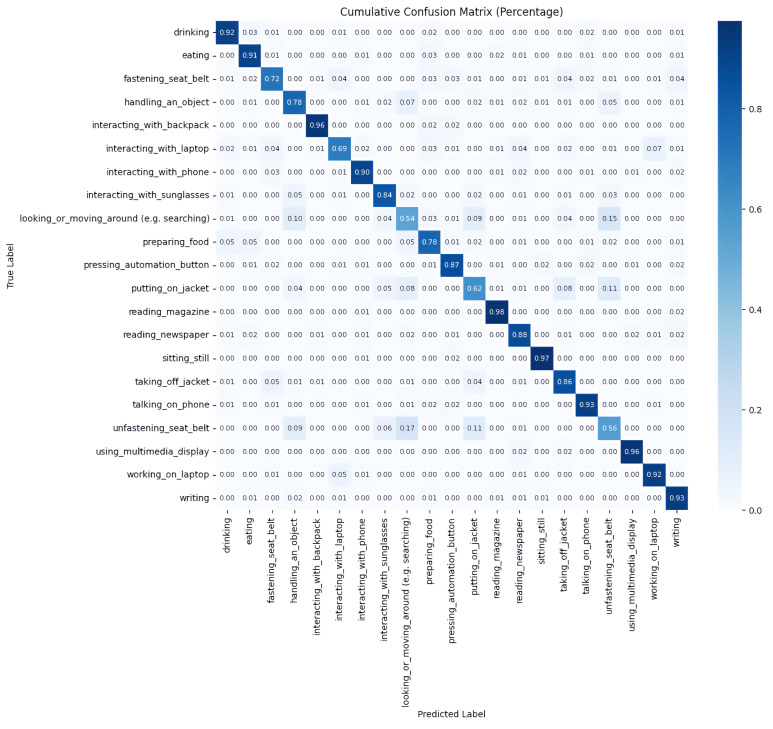
Cumulative confusion matrix for the proposed model of SAR.

**Figure 14 sensors-24-06604-f014:**
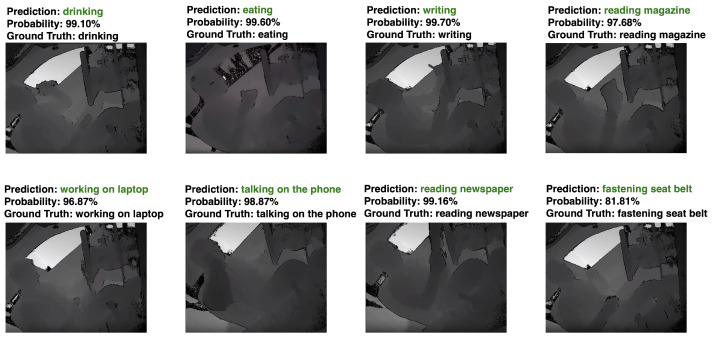
The predictions and prediction probabilities of the proposed model for SAR for some sample clips. The green font indicates the correct prediction of our proposed method.

**Figure 15 sensors-24-06604-f015:**
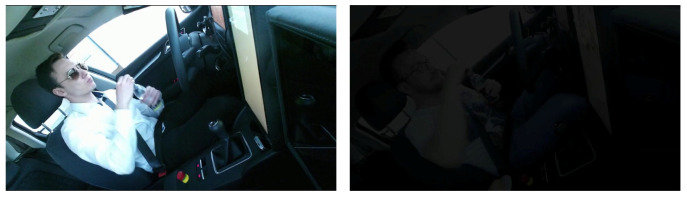
The left image is an RGB video clip in light condition and the right image is an RGB video clip in dark condition.

**Figure 16 sensors-24-06604-f016:**
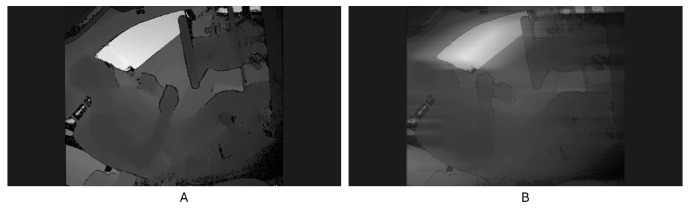
Image (**A**) is a frame from the original videos, and Image (**B**) is the frame from the video after adding ‘camera shake’ and ‘motion blur’ effects.

**Table 1 sensors-24-06604-t001:** Summary of video data parameters.

Parameter	Value
Number of action categories	21
Average videos per action category	123
Average number of frames per video	95
Resolution of the video	512×424
Average frames per second (FPS) of the video	30

**Table 2 sensors-24-06604-t002:** Specifications of Microsoft Kinect for Xbox One.

Specifications	Microsoft Kinect for Xbox One
Depth sensor type	Time of Flight (ToF)
RGB camera resolution	1920 × 1080, 30 fps
IR camera resolution	512 × 424, 30 fps
Field of view of RGB camera	84.1° × 53.8°
Field of view of depth camera	70.6° × 60°
Operative measuring range	0.5 m–4.5 m
Skeleton joints defined	25 joints
Maximum skeletal tracking	6

**Table 3 sensors-24-06604-t003:** The configuration of the experimental environment.

Name	Parameter
GPU	Quadro RTX 5000, 16,384 MB
System	Windows 11
Operating memory	16 GB
Environment configuration	Python-3.8.19 tensorflow-2.13.0

**Table 4 sensors-24-06604-t004:** The hyperparameters used during the training.

Name	Value
Patience	15
Loss function	categorical cross-entropy
Optimizer	Adam
Metrics	accuracy
Epochs	60
Batch size	4
Shuffle	True
Validation split	20%

**Table 5 sensors-24-06604-t005:** K-fold cross validation.

Fold-n	Overall Average Test Loss	Overall Average Accuracy (%)
Fold-1	0.57	83.40
Fold-2	0.82	79.05
Fold-3	0.59	83.16
Fold-4	0.51	86.07
Fold-5	0.44	87.73
**Average**	**0.59**	**83.88**

**Table 6 sensors-24-06604-t006:** Comparison results of SAR accuracy of the dataset before and after DA and SR.

Action	Original Data (%)	After DA (%)	After SR (%)
**Overall**	**69.27**	**74.36**	**83.88**
Drinking	66.67	78.72	91.53
Eating	90.91	88.64	91.06
Fastening Seat Belt	53.12	51.35	72.17
Interacting with Phone	83.72	94.74	90.27
Handling an Object	54.78	43.71	77.97
- Fetching an Object	61.82	-	-
- Placing an Object	47.73	-	-
Interacting with Backpack	60.72	81.58	95.92
- Opening Backpack	50.00	-	-
- Putting Laptop into Backpack	71.43	-	-
Interacting with Laptop	39.17	42.86	69.39
- Closing Laptop	55.56	-	-
- Opening Laptop	23.08	-	-
Interacting with Sunglasses	20.71	75.00	83.61
- Putting on Sunglasses	12.00	-	-
- Taking off Sunglasses	29.41	-	-
Preparing Food	60.00	70.00	78.30
Pressing Automation Button	78.57	92.11	86.89
Putting on Jacket	46.43	57.14	61.68
Reading Magazine	97.00	100.00	97.52
Reading Newspaper	92.68	93.18	87.60
Taking off Jacket	70.00	65.52	86.09
Talking on Phone	91.67	78.95	92.56
Unfastening Seat Belt	39.13	46.15	56.00
Using Multimedia Display	92.31	86.21	95.93
Working on Laptop	92.11	87.80	92.44
Writing	79.31	80.00	93.50
Looking or Moving Around (e.g., searching)	17.65	50.00	53.57
Sitting Still	100.00	97.87	97.32

**Table 7 sensors-24-06604-t007:** Comparison results of SAR accuracy of models with and without SEAM.

Model	Accuracy
CNN+Bi-LSTM	72.12%
**CNN-SEAM+Bi-LSTM (ours)**	**83.88%**

**Table 8 sensors-24-06604-t008:** Comparison results of SAR accuracy of LSTM and Bi-LSTM models with different hidden units.

Model	64 Units	128 Units	256 Units	512 Units
One-layer LSTM	78.36%	78.98%	81.47%	78.48%
Two-layer LSTM	77.99%	80.11 %	78.48%	77.38%
Bi-LSTM	80.59%	82.29%	**83.88%**	83.37%

**Table 9 sensors-24-06604-t009:** Comparison results of SAR accuracy of models with different CNN layers.

Model	Number of CNN Layers	Accuracy
CNN-SEAM+Bi-LSTM	3 layers	80.92%
	4 layers	**83.88%**
	5 layers	76.56%

**Table 10 sensors-24-06604-t010:** Comparison results of SAR accuracy of state-of-the-art methods on benchmark Drive&Act dataset.

Model	Year	Accuracy (%)
ConvLSTM [50]	2015	56.87
LRCN [51]	2015	66.23
CNN+Deep BiLSTM [8]	2017	69.43
SE-LRCN [52]	2018	70.12
I3D [9]	2019	60.97
TransDARC [29]	2021	8.01
st-mlp [53]	2022	22.58
Distillation-Based Neural Architecture [31]	2022	65.69
Mobilenet [32]	2023	39.77
Quantized Distillation [32]	2023	48.17
Knowledge Distillation [32]	2023	48.56
**CNN-SEAM+Bi-LSTM (ours)**	-	**83.88**

**Table 11 sensors-24-06604-t011:** Comparison results of SAR accuracy of the dataset under different data types.

Action	Depth Videos (%)	RGB Videos (%)	RGB Videos (Dark) (%)
**Overall**	**83.88**	**65.87**	**58.36**
Drinking	91.53	48.28	55.17
Eating	91.06	86.36	81.82
Fastening Seat Belt	72.17	40.74	29.63
Handling an Object	77.97	46.05	42.49
Interacting with Phone	90.27	89.80	89.80
Interacting with Backpack	95.92	62.07	62.07
Interacting with Laptop	69.39	48.28	34.48
Interacting with Sunglasses	83.61	27.59	6.90
Preparing Food	78.30	50.00	35.71
Pressing Automation Button	86.89	89.66	86.21
Putting on Jacket	61.68	71.76	69.70
Reading Magazine	97.52	93.75	100.00
Reading Newspaper	87.60	90.24	75.61
Taking off Jacket	86.09	25.00	31.25
Talking on Phone	92.56	96.97	66.67
Unfastening Seat Belt	56.00	41.18	23.53
Using Multimedia Display	95.93	93.94	78.79
Working on Laptop	92.44	100.00	85.37
Writing	93.50	70.97	70.97
Looking or Moving Around (e.g., searching)	53.57	34.78	17.39
Sitting Still	97.32	95.74	97.87

**Table 12 sensors-24-06604-t012:** Comparison results on the three additional datasets.

Model	Datasets	Number of Categories	Accuracy
CNN-SEAM+Bi-LSTM	KTH [54]	6	65.60%
	UCF50 [55]	50	89.24%
	UCF101 [56]	101	85.06%

**Table 13 sensors-24-06604-t013:** A comparison of the results before and after adding the shaking effect to the depth videos.

Model	Datasets	Accuracy
CNN-SEAM+Bi-LSTM	Before adding shaking effects	83.88%
	After adding shaking effects	81.34%

## Data Availability

Our research uses a publicly available dataset: Drive&Act (https://driveandact.com/).

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
