# Peer review of "Depth Video-Based Secondary Action Recognition in Vehicles via Convolutional Neural Network and Bidirectional Long Short-Term Memory with Spatial Enhanced Attention Mechanism"

_sensors, 2024, doi:10.3390/s24206604_

Round 1

Reviewer 1 Report

Comments and Suggestions for Authors The paper introduces a comprehensive framework that integrates Convolutional Neural Network (CNN) and Bidirectional Long Short-Term Memory (Bi-LSTM) networks, enhanced by a Spatial Enhanced Attention Mechanism (SEAM). This combination significantly improves the model's ability to recognize secondary actions in vehicles, which are crucial for road safety and minimizing distractions while drivingSuggestions for Improvement
  1. The current model focuses on recognizing actions of a single person (the driver). Future work could include recognizing actions of multiple individuals in the vehicle, such as passengers, to provide a more comprehensive understanding of in-car dynamics.

  2. Enhancing the model to support real-time action recognition could significantly improve its applicability in real-world scenarios, such as Advanced Driving Assistance Systems (ADAS). This would require optimizing the model for faster inference times without sacrificing accuracy.

  3. To improve the model's robustness, it could benefit from training on a more diverse dataset that includes various driving conditions, different vehicle types, and a wider range of secondary actions. This would help the model generalize better to unseen scenarios.

  4. Implementing a feedback loop where users can report inaccuracies or suggest improvements could help refine the model over time. This could be particularly useful in adapting the model to specific user behaviors or preferences.

  5. Combining depth video data with other sensor inputs, such as audio or environmental sensors, could provide a richer context for action recognition. This multimodal approach could enhance the model's understanding of the driver's state and intentions, leading to better safety measures.

Reviewer 2 Report

Comments and Suggestions for Authors

THis is an interesting paper proposing appliocation of depth-video instead of traditional (RGB) video for detecting secondary driving actions. The paper is generally well-written however, it misses some technical details.

For example:

- What is the resolution of used depth-images?

- What devices (sensors) can provide it (besides XBox Keenetic)?

- What hardware is needed to be able to detect secondary actions "on the fly" including super-scaling and secondary action classification?

- How does shaking (which is a very common problem during driving) affect the results?

- Are there any other limitaions of the approach? The discussion section is completely missing.

Round 2

Reviewer 1 Report

Comments and Suggestions for Authors

The authors considered my comments and suggestions. Good luck.